The systematics of the Mongolepidida (Chondrichthyes) and the Ordovician origins of the clade

Andreev Plamen 1 p.andreev@bham.ac.uk
Coates Michael I. 2
Karatajūtė-Talimaa Valentina 3
Shelton Richard M. 4
Cooper Paul R. 4
Wang Nian-Zhong 5
Sansom Ivan J. 1 i.j.sansom@bham.ac.uk
1 School of Geography, Earth and Environmental Sciences, University of Birmingham , Birmingham , United Kingdom
2 Department of Organismal Biology and Anatomy, University of Chicago , Chicago , United States
3 Department of Geology and Mineralogy , Vilnius University , Vilnius , Lithuania
4 School of Dentistry, University of Birmingham , Birmingham , United Kingdom
5 Institute of Vertebrate Paleontology and Paleoanthropology, Chinese Academy of Sciences , Beijing , China
De Baets Kenneth
Electronic publication date: 2016 Jun 16
Publication date: 2016
Volume: 4
Electronic Location ID: e1850
Received 2015 Sep 23; Accepted 2016 Mar 5
Copyright: ©2016 Andreev et al.
Copyright year: 2016
Copyright holder: Andreev et al.
License: This is an open access article distributed under the terms of the Creative Commons Attribution License, which permits unrestricted use, distribution, reproduction and adaptation in any medium and for any purpose provided that it is properly attributed. For attribution, the original author(s), title, publication source (PeerJ) and either DOI or URL of the article must be cited.
License URL: https://creativecommons.org/licenses/by/4.0/

Keywords: Solinalepis gen. nov., Scales, Odontocomplex, Morphogenesis, Ordovician, Mongolepids

Funding: University of Birmingham, School of Geography, Earth and Environmental Sciences Research Council GR3/8543, NER/B/S/2000/0028 Small Grant Awards AGM 2011 of the Palaeontogical Association: Sylvester-Bradley Award P Andreev Doctoral Studentship from the University of Birmingham, School of Geography, Earth and Environmental Sciences Research Council Grants: GR3/8543 and NER/B/S/2000/0028. Small Grant Awards AGM 2011 of the Palaeontogical Association: Sylvester-Bradley Award. The funders had no role in study design, data collection and analysis, decision to publish, or preparation of the manuscript.

==============================
The Mongolepidida is an Order of putative early chondrichthyan fish, originally erected to unite taxa from the Lower Silurian of Mongolia. The present study reassesses mongolepid systematics through the examination of the developmental, histological and morphological characteristics of scale-based specimens from the Upper Ordovician Harding Sandstone (Colorado, USA) and the Upper Llandovery–Lower Wenlock Yimugantawu (Tarim Basin, China), Xiushan (Guizhou Province, China) and Chargat (north-western Mongolia) Formations. The inclusion of the Mongolepidida within the Class Chondrichthyes is supported on the basis of a suite of scale attributes (areal odontode deposition, linear odontocomplex structure and lack of enamel, cancellous bone and hard-tissue resorption) shared with traditionally recognized chondrichthyans (euchondrichthyans, e.g., ctenacanthiforms). The mongolepid dermal skeleton exhibits a rare type of atubular dentine (lamellin) that is regarded as one of the diagnostic features of the Order within crown gnathostomes. The previously erected Mongolepididae and Shiqianolepidae families are revised, differentiated by scale-base histology and expanded to include the genera Rongolepisand Xinjiangichthys, respectively. A newly described mongolepid species (Solinalepis levis gen. et sp. nov.) from the Ordovician of North America is treated as family incertae sedis, as it possesses a type of basal bone tissue (acellular and vascular) that has yet to be documented in other mongolepids. This study extends the stratigraphic and palaeogeographic range of Mongolepidida and adds further evidence for an early diversification of the Chondrichthyes in the Ordovician Period, 50 million years prior to the first recorded appearance of euchondrichthyan teeth in the Lower Devonian.

Introduction

Middle Ordovician to upper Silurian strata have yielded a number of isolated scale remains that have been assigned to the chondrichthyans with varying degrees of confidence. This 50-million-year record pre-dates the first appearance of teeth and articulated skeletons (Leonodus and Celtiberina Botella, Donoghue & Martínez-Pérez, 2009; Doliodus Miller, Cloutier & Turner, 2003; Maisey, Miller & Turner, 2009 and Antarctilamna Young, 1982) of traditionally recognized chondrichthyans (euchondrichthyans sensu Pradel et al., 2014), as well as body fossils of acanthodian-grade stem chondrichthyans (Brazeau & Friedman, 2015 and references therein). These, largely microscopic, remains include the elegestolepids (Karatajūtė-Talimaa, 1973; Žigaitė & Karatajūtė-Talimaa, 2008; Andreev et al., in press), sinacanthids (Zhu, 1998; Sansom, Wang & Smith, 2005; Zeng, 1988), taxa such as Tezakia and Canyonlepis from the Ordovician of North America (Sansom, Smith & Smith, 1996; Andreev et al., 2015), Tantalepis (Sansom et al., 2012), Kannathalepis (Märss & Gagnier, 2001) and Pilolepis (Thorsteinsson, 1973), and, perhaps the most widely distributed and diverse collection of what Ørvig and Bendix-Almgreen, quoted in Karatajūtė-Talimaa (1995), referred to as ‘praechondrichthyes,’ the mongolepids (Karatajūtė-Talimaa et al., 1990; Karatajūtė-Talimaa & Predtechenskyj, 1995; Sansom, Aldridge & Smith, 2000). It is the latter which this work concentrates on, re-assessing and re-defining previously described members of the Mongolepidida, and describing a new taxon that extends the range of the Order into the Ordovician, adding further evidence for a diversification of early chondrichthyans as part of the Great Ordovician Biodiversification Event that encompasses a wide variety of taxa, both invertebrate (e.g., Webby, Paris & Droser, 2004; Servais et al., 2010) and vertebrate (Sansom, Smith & Smith, 2001; Turner, Blieck & Nowlan, 2004).

Previous work on mongolepids

Mongolepids were first described by Karatajūtė-Talimaa et al. (1990) and Karatajūtė-Talimaa (1995) from the Chargat Formation (Upper Llandovery–Lower Wenlock) in north-western Mongolia, together with a diverse assemblage of early vertebrates including pteraspidomorphs (V Karatajūtė-Talimaa, 2013, unpublished data), thelodonts (Žigaitė, 2013; Žigaitė 2004; Žigaitė, Karatajūtė-Talimaa & Blieck, 2011), acanthodians (Karatajūtė-Talimaa & Smith, 2003) and elegestolepids. The first erected species, Mongolepis rozmanae, was subsequently added to with the description of Teslepis jucunda Karatajūtė-Talimaa & Novitskaya (1992) and Sodolepis lucens Karatajūtė-Talimaa & Novitskaya (1997), also from the Chargat Formation. Recently the stratigraphic ranges of Mongolepis and Teslepis have been extended to include Aeronian (Middle Llandovery) and Sheinwoodian (Lower Wenlock) sedimentary sequences from Altai and Tuva (Sennikov et al., 2015). Shiqianolepis hollandi from the Xiushan Formation (Telychian) of south China was also placed within the Order by Sansom, Aldridge & Smith (2000), although a new Family, the Shiqianolepidae, was erected based upon an interpretation of the scale growth patterns within mongolepids. Additional material from the upper Llandovery of the Tarim Basin (Xinjiang Uygyr Autonomous Region, north-west China) is also referable to the group (NZ Wang, 2011, unpublished data). Thus, to date, the distribution of mongolepids has been limited to a very narrow time frame (Llandovery–Wenlock) and is concentrated within the Mongol-Tuva, Altai, South China and Tarim tectonic blocks. The taxonomic placement of the group has been greatly hampered by the absence of articulated specimens that exhibit any anatomical detail of the mongolepid bodyplan (Karatajūtė-Talimaa et al., 1990; Karatajūtė-Talimaa, 1995).

Materials and Methods

All examined material consists of isolated scales extracted by petroleum ether or acetic acid disaggregation of rock samples from the Sandbian Harding Sandstone of central Colorado, USA, the Upper Llandovery–Lower Wenlock Chargat Formation of north-western Mongolia, the lower and upper members of the Telychian Yimugantawu Formation of Xinjiang (Tarim Basin, China) and the lower Member of the Telychian Xiushan Formation (Guizhou Province, China).

Scale morphology was documented using the JEOL JSM-6060 and Zeiss EVO LS scanning electron microscopes at the School of Dentistry of the University of Birmingham, UK. Prior to imaging specimens were sputter-coated with gold/palladium alloy.

For the purpose of studying scale histology and internal structure, doubly polished thin sections of scales were examined with Nomarski differential interference contrast microscopy (using a ‘Zeiss Axioskop Pol’ polarization microscope) and scanning electron microscopy (using a JEOL JSM-6060 SEM at the School of Dentistry, University of Birmingham, UK).

Scale examination with X-ray radiation was performed with the SkyScan 1172 microtomography scanner at the School of Dentistry, University of Birmingham, UK. The acquired microradiographs (tomographic projections) were taken at 0.3° intervals over a 180° rotation cycle at exposure times of 400 ms, using a 0.5 mm thick X-ray attenuating Al filter. These image data were processed with the SkyScan NRecon reconstruction software for the purpose of generating sets of microtomograms that were converted into volume renderings in Amira 5.4 3D analysis software.

Figured specimens are housed in the Lapworth Museum of Geology, University of Birmingham, UK (BU prefix), the Nanjing Institute of Geology and Palaeontology, Chinese Academy of Sciences, Nanjing, China (NIGP prefix) and the Institute of Vertebrate Paleontology and Paleoanthropology, Chinese Academy of Sciences, Beijing, China (IVPP V prefix). The examined non-figured scales have not been given collection numbers. Additional mongolepid material, alluded to but not described in this work (referred to as unpublished data above), from the Tarim Basin of China is housed in and registered at the Institute of Vertebrate Paleontology and Paleoanthropology, Chinese Academy of Sciences.

The electronic version of this article in Portable Document Format (PDF) will represent a published work according to the International Commission on Zoological Nomenclature (ICZN), and hence the new names contained in the electronic version are effectively published under that Code from the electronic edition alone. This published work and the nomenclatural acts it contains have been registered in ZooBank, the online registration system for the ICZN. The ZooBank LSIDs (Life Science Identifiers) can be resolved and the associated information viewed through any standard web browser by appending the LSID to the prefix http://zoobank.org/. The LSID for this publication is: urn:lsid:zoobank.org:pub:3C24AE11-1F12-4B16-B04D-480CA204CCEA. The online version of this work is archived and available from the following digital repositories: PeerJ, PubMed Central and CLOCKSS.

Definitions of terms

The interpretations of the terms (Fig. 1) employed in the descriptions of fossil scales follow Andreev et al. (2015). The rationale behind this is to improve identification of homologous scale structures across taxa by introducing a standardized terminology.

Figure 1 Principle morphological features of scales.

Line drawing of a Mongolepis scale (BU5296) from the Chargat Formation of north-western Mongolia in lateral view.

Results

Systematic palaeontology

Class CHONDRICHTHYES Huxley, 1880	
Order MONGOLEPIDIDA Karatajūtė-Talimaa et al., 1990	

Included families

Mongolepididae Karatajūtė-Talimaa et al., 1990

Shiqianolepidae Sansom, Aldridge & Smith, 2000

Emended diagnosis

Polyodontode growing scale crowns formed by multiple antero-posteriorly oriented primary odontocomplex rows. Odontode size within each row increases gradually towards the posterior of the scale. Individual odontodes formed exclusively of inotropically and spheritically mineralised atubular, acellular dentine (lamellin).

Remarks

The current study has determined scale crown growth (sensu Reif, 1978) to be a characteristic shared by all mongolepid taxa (see Discussion for details), contrary to previous interpretations of synchronomorial development of scale odontodes in Mongolian mongolepid species (Karatajūtė-Talimaa et al., 1990; Karatajūtė-Talimaa & Novitskaya, 1992; Karatajūtė-Talimaa & Novitskaya, 1997). Under the revised definition of the Order, the Mongolepidida retains the Families Mongolepididae (Karatajūtė-Talimaa et al., 1990) and Shiqianolepidae (Sansom, Aldridge & Smith, 2000), yet contra Sansom, Aldridge & Smith (2000) these are newly differentiated on the basis of base histology (see below) and are expanded to also include the genera Rongolepis Sansom, Aldridge & Smith, 2000 and Xinjiangichthys Wang et al., 1998, respectively. Solinalepis levis gen. et sp. nov. is also added to the Order, but placed within incertae sedis at Family-grade due to the absence of clearly defined characters at this taxonomic level.

Family MONGOLEPIDIDAE Karatajūtė-Talimaa et al., 1990

Included genera

Mongolepis Karatajūtė-Talimaa et al., 1990	
Teslepis Karatajūtė-Talimaa & Novitskaya, 1992	
Sodolepis Karatajūtė-Talimaa & Novitskaya, 1997	
Rongolepis Sansom, Aldridge & Smith, 2000	

Emended diagnosis

Mongolepids possessing bulging scale bases composed of acellular bone tissue with cross-ply architecture.

Figure 2 Character distribution within Mongolepidida.

Cladogram based on a yet-to-be-published scale-based phylogeny of early chondrichthyans by P Andreev, M Coates & I Samson (2014, unpublished data). Portion of a majority-rule consensus tree generated in TNT version 1.1 (Goloboff, Farris & Nixon, 2008) using a data matrix of 68 equally weighted scale-based characters (53 original and 15 revised/adopted) and 49 Palaeozoic jawed-gnathostome taxa.

Remarks

Scale-derived phylogenetic data (Fig. 2; Andreev et al., in press) identify two monophyletic groups inside Mongolepidida distinguished by differences in the bone histology and morphology of the scale base. These substitute the scale-crown developmental characteristics that have been used previously by Sansom, Aldridge & Smith (2000) to establish the Family structure of the Mongolepidida.

Genus MONGOLEPIS Karatajūtė-Talimaa et al., 1990

Type and only species

Mongolepis rozmanae Karatajūtė-Talimaa et al. (1990), from the Chargat Formation, Salhit regional Stage (Upper Llandovery–Lower Wenlock) of north-western Mongolia. Non-figured M. rozmanae and M. sp. specimens have been reported (Sennikov et al., 2015) from the Aeronian (Middle Llandovery) Sadra section (Gornaya Shoriya, Altai Republic, Russia) and the Sheinwoodian (Lower Wenlock) Upper Tarkhata Subformation (Charygka horizon, Gorny Altai, Altai Republic, Russia) and Baytal Formation (Pichishui Horizon, Tuva Republic, Russia).

Diagnosis

As for the type species.

MONGOLEPIS ROZMANAE Karatajūtė-Talimaa et al., 1990 (Figs. 1, 3A–3D, 6A–6E, 8A–8C, 9D)

1990 Mongolepis rozmanae Karatajūtė-Talimaa, Novitskaya, Rozman & Sodov, Figs. 2–5, pl. IX	
1992 Mongolepis rozmanae Karatajūtė-Talimaa & Novitskaya, Fig. 2ж, 3.	
1995 Mongolepis rozmanae Karatajūtė-Talimaa, Fig. 1.	
1998 Mongolepis rozmanae Karatajūtė-Talimaa, Figs. 11 and 20.	

Emended diagnosis

Mongolepidids (pertaining to Mongolepididae) possessing large scales (up to over 3 mm), constricted along their anterior margin, containing a large number of primary odontocomplex rows (up to 50+) with long, sigmoidal odontodes. Inter-odontocomplex spaces divided into pore-like compartments by short, transverse struts. Bulbous base with a prominent crescent-shaped anterior platform that forms below the level of the crown surface and extends laterally into two spine-shaped processes.

Holotype

An ontogenetically mature scale (LGI M-1-031) deposited in collection LGI M-1 of the Lithuanian Geological Survey, Vilnius (Karatajūtė-Talimaa et al., 1990).

Referred material

Hundreds of isolated scales from the type locality (from samples P-16/3 and ЦГЭ N1009). Non-figured specimens examined for this study are stored in the microvertebrate research collection of the Lapworth Museum of Geology, University of Birmingham, UK.

Figure 3 Scale morphology of Mongolepididae.

(A–C) Mongolepis rozmanae scale BU5296 (Chargat Formation, north-western Mongolia) in (A) anterior (B) lateral, (C) and basal aspect and a M. rozmanae scale in (D) crown view (BU5351, Chargat Formation, north-western Mongolia); (E, G) Teslepis jucunda BU5322 (Chargat Formation, north-western Mongolia) in (E) crown and (G) basal view and a T. jucunda scale (BU5352, Chargat Formation, north-western Mongolia) in an (F) antero-lateral view; (H–J) Sodolepis lucens scales (Chargat Formation, north-western Mongolia) in (H) lateral (BU5305), crown (BU5304) and (J) basal (BU5355) views; (K–M) Rongolepis cosmetica scale BU5303 (Xiushan Formation, south China) in (K) crown, (L) lateral and (M) basal views;. Volume renderings, (A–C), (H) and (K–M); SEM micrographs, (D–G) and (I, J). Crown and base foramina indicated by arrows and arrowheads respectively. Anterior to the left in (B), (H), (L) and bottom in (A–G), (H–K), (M). Scale bar equals 500 µm in (D, I, J), 400 µm in (A–C), 300 µm in (H, K) and 200 µm in (E–G, L, M).

Description

Morphology

Primary odontodes from the same position in the crown are of equal size irrespective of scale dimensions. The number of odontocomplex rows changes with the proportions of the crown and its size, with scales of up to 2 mm in length usually possessing less than 20 odontocomplexes, whereas in larger specimens their number varies from 20 to c. 35.

Primary odontodes exhibit posteriorly curved profiles and an incremental increase in length towards the posterior of the scale (Figs. 6A, 6B and 9D). This creates a significant height difference (over five fold in medial odontocomplexes) between the anterior- and the posterior-most elements of primary odontocomplexes, whilst odontode thickness remains relatively constant at c. 50 µm (Figs. 6A, 6B and 9D). The crown surface profile is planar (Figs. 3A, 3B and 3D) due to a gradual decrease in the angle of odontode curvature towards the posterior of the scale, accompanied by sloping of the crown/base contact surface (Figs. 6A and 9D).

Figure 4 Scale morphology of Shiqianolepidae.

(A–C) Shiqianolepis hollandi scales (Xiushan Formation, south China) in (A) lateral (NIGP 130307), (B) crown (NIGP 130309) and (C) postero-basal (NIGP 130307) views; (D–F) Xinjiangichthys pluridentatus scale IVPP V X2 (Yimugantawu Formation, north-western China) in (D) anterior, (E) posterior and (F) antero-lateral views. All images volume renderings except (B). Crown foramina indicated by arrows. Anterior to the left in (A), to the right in (F) and bottom in (B). Scale bar equals 300 µm in (A, B) and 200 µm in (C–F).

In scales larger than 1 mm, secondary odontodes are developed to a varying extent along the anterior margin of the crown (Figs. 3A, 3B and 3D). These are arranged into rows and are undivided by inter-odontode spaces (Figs. 3A, 3B and 3D). In common with the main crown odontodes, the secondary odontodes are posteriorly arched elements that demonstrate an unidirectional increase in length (Figs. 6A–6B and 9D); the latter being expressed towards the anterior end of the scale.

Figure 5 SEM micrographs of Solinalepis levis gen. et sp. nov. scales from the Upper Ordovician Harding Sandstone of Colorado, USA.

(A–C) tessera-like head scales in (A, B) crown (BU5307, BU5308) and (C) lateral (BU5309) views; (D) bulbous head scale (BU5312) in lateral view; (E–I) polygonal trunk scales, (E) holotype (BU5310) in anterior view, (F) BU5345 in crown, (G) corono-lateral and (H) partial posterior views, (I) BU5313 in basal view; (J–L) lanceolate trunk scales in (J) anterior (BU5314), (K) lateral (BU5315) and (L) posterior (BU5311) views. Base foramina indicated by arrowheads. Anterior to the left in (G) and (K). Scale bar equals 300 µm in (A, B), 200 µm in (C), 100 µm in (D–G, I–L), and 50 µm in (H).

The scale bases are bulbous structures (Fig. 3A–3C) that reach their maximum thickness directly under the anterior apex of the crown. To the posterior, the majority of scale bases display a pitted lower-base surface produced by series of canal openings (Figs. 3B and 3C).

Figure 6 Scale histology of Mongolian and Chinese mongolepids.

(A) medial longitudinal section of a Mongolepis rozmanae scale (BU5297; Chargat Formation, north-western Mongolia); (B) detail of (A) depicting primary and secondary odontodes at the anterior crown margin; (C) primary odontode lamellin microstructure in a longitudinally sectioned Mongolepis rozmanae scale (BU5298; Chargat Formation, north-western Mongolia), etched for 10 min in 0.5% orthophosphoric acid; (D) basal bone microstructure of a longitudinally sectioned Mongolepis rozmanae scale (BU5354; Chargat Formation, north-western Mongolia) etched for 10 min in 0.5% orthophosphoric acid; (E) detail of BU5354 depicting the bone tissue of the anterior basal platform; (F) medial longitudinal section of a Teslepis jucunda scale (BU5324; Chargat Formation, north-western Mongolia); (G) lamellin architecture of two odontodes in a longitudinally sectioned Sodolepis lucens scale (BU5306; Chargat Formation, north-western Mongolia) etched for 10 min in 0.5% orthophosphoric acid; (H) basal bone microstructure in BU5306 at the anterior projection of the base; (I), sagittal longitudinal section of a Sodolepis lucens scale (BU5344; Chargat Formation, north-western Mongolia); (J) anterior third of BU5306 showing the contact between the globular crown dentine and the underlying basal bone; (K) sagittal longitudinal section of a Rongolepis cosmetica scale (NIGP 130328; Xiushan Formation, south China); (L) detail of NIGP 130328 showing the mid third of the scale crown; (M) Xinjiangichthys pluridentatus scale (IVPP V X1; Yimugantawu Formation, north-western China) in longitudinal section; (N) sagittal longitudinal section of a Shiqianolepis hollandi trunk scale (NIGP 130312; Xiushan Formation, south China). Nomarski differential interference contrast optics micrographs, (A), (B), (D), (F), (G), (I) and (K–N); SEM micrographs, (C), (E), (H) and (J). Anterior towards the left in (A–J, L) and towards the right in (K), (M) and (N). Abbreviations: gb, globular dentine; lb, lamellar bone; red dotted lines, contact surfaces between primary and secondary odontodes; white dotted lines, border between globular dentine and basal bone; white dashed line, contact surfaces between primary odontodes in Rongolepis. Asterisks mark bone layers with fibre orientation parallel to the section axis. Scale bar equals 400 µm in (A), 100 µm in (B, G, H, M), 20 µm in (C), 200 µm in (D, F, K, N), 50 µm in (E, J, L), and 300 µm in (I).

Histology

Scale odontodes are composed of atubular dentine (Fig. 6A–6C) for which Karatajūtė-Talimaa et al. (1990) used the term lamellin (first introduced by Bolshakova & Ulitina, 1985). Within individual odontodes, the lamellin displays two histologically distinct regions—a peripheral (10–20 µm thick) lamellar zone and an inner region dominated by mineralised spherites united within Liesegang waves (Fig. 6C; for a definition refer to (Ørvig, 1951)). The diameter of the calcospherites changes randomly but rarely exceeds 15 µm.

Primary odontode pulps are either closed off or can be greatly constricted by dentine infill (Figs. 6A and 8C) yet remaining open at their lower end, from which emerges a pair of short (c. 15 µm) horizontal canals that connect the pulp cavity to the odontode surface (Fig. 8C, C1). The foramina of these canals face either the inter-odontocomplex spaces or, in marginal odontodes, are exposed at the periphery of the crown (Fig. 3A).

In a similar manner to primary odontocomplexes, the pulps of secondary odontodes are substantially constricted by dentine deposition, but they lack the network of horizontal canals (Figs. 3A, 3B and 8C) developed inside the rest of the crown.

The scale base consists of acellular bone characterized by a succession of convex-down growth lamellae (up to 150 µm thick; Figs. 6A, 6D and 9D) that increase in extent towards the lower portion of the tissue. Secondary lamination is evident within these primary depositional structures and is produced by intrinsic mineralised fibres (sensu Ørvig, 1966) of c. 2 µm diameter, which likewise demarcate the boundary surfaces of primary lamellae (Fig. 6D). The basal bone also contains elaborately organised extrinsic crystalline fibres (sensu Ørvig, 1966) of c. 2 µm diameter (Figs. 6A and 6E), which have the appearance of hollow cylindrical rods (Fig. 5E). These are grouped into layers oriented obliquely with respect to one another (Figs. 6A, 6E and 9D), that propagate through the tissue. The layers exhibit straight to upwardly arching profiles and thickness of c. 50–70 µm (Figs. 6A, 6D, 6E and 9D).

Figure 7 Histology of Solinalepis levis gen. et sp. nov. scales.

(A) thin-sectioned head scale (BU5317) from the Harding Sandstone, Colorado, USA; (B) transverse section of a Solinalepis levis gen. et sp. nov. trunk scale (BU5316) from the Harding Sandstone, Colorado, USA. Scale bar equals 200 µm in (A) and 100 µm in (B).

The base houses a vascular system represented by curved (both anteriorly and posteriorly) large-calibre vertical canals (c. 100 µm; Figs. 8A and 8B) that are split at their upper end into two or more rami, each merging with one of the primary odontode pulps. Conversely, the secondary odontode pulps are not connected to the canal system of the base.

Remarks

In contrast to earlier work on Mongolepis (Karatajūtė-Talimaa et al., 1990; Karatajūtė-Talimaa, 1998), the present study reinterprets the pattern of scale ontogenesis of the genus. Recorded size differences between Mongolepis scales have been used by previous authors (Karatajūtė-Talimaa et al., 1990; Karatajūtė-Talimaa, 1998) to identify four distinct ontogenetic stages in the development of the scale cover. They have suggested synchronomorial crown growth succeeded by incremental deposition of basal bone to characterise the scale morphogenesis of Mongolepis, with scales of ever-increasing crown size and base thickness assumed to be added at each stage of scale cover ontogeny. A re-examination of Mongolepis specimens has revealed the presence of bases across the spectrum of documented scale sizes. Furthermore, specimens in the sub-millimetre size category, corresponding to the papillary and juvenile scales of Karatajūtė-Talimaa et al. (1990), possess bases that are proportionally as thick as those of larger scales. Thus, scales interpreted as being composed exclusively of odontodes (Karatajūtė-Talimaa, 1998, Fig. 11A2, E) were related to specimens where the bases had been abraded away. This new morphological evidence supports incremental and mutually synchronous deposition of Mongolepis crown and base scale components. The odontocomplex structure and base depositional lamellae of Mongolepis scales are similarly identified in all mongolepid genera and indicate that cyclomorial scale growth, achieved via sequential areal addition of odontodes (sensu Sansom, Aldridge & Smith (2000), originally defined by Stensiö (1961)), is a characteristic of the Mongolepidida (see Discussion for details).

Genus TESLEPIS Karatajūtė-Talimaa & Novitskaya, 1992

Type and only species

Teslepis jucunda (Karatajūtė-Talimaa & Novitskaya, 1992), from the Chargat Formation (Salhit regional Stage, Upper Llandovery–Lower Wenlock) of north-western Mongolia. Non-figured T. jucunda specimens have been reported (Sennikov et al., 2015) from the Aeronian (Middle Llandovery) Sadra section (Gornaya Shoriya, Altai Republic, Russia) and the Sheinwoodian (Lower Wenlock) Upper Tarkhata Subformation (Charygka horizon, Gorny Altai, Altai Republic, Russia).

Diagnosis

As for the type species.

TESLEPIS JUCUNDA Karatajūtė-Talimaa & Novitskaya, 1992 (Figs. 3E–3G, 6F, 8D, 9A)

1992 Teslepis jucunda Karatajūtė-Talimaa & Novitskaya, Figs. 1, 2A–E, 3, 4, pl. V Figs. 1–8.	
1992 Teslepis sp. Karatajūtė-Talimaa & Novitskaya, pl. V Fig. 9.	
1998 Teslepis jucunda Karatajūtė-Talimaa, Fig. 19.	

Emended diagnosis

Mongolepidids with small scales whose odontocomplex number increases with scale size. Non-odontode atubular globular dentine developed at the anterior and lateral crown margins. Scale base extended into an antero-basally directed conical projection.

Holotype

An ontogenetically mature scale (LGI M-1-077) deposited in collection LGI M-1 of the Lithuanian Geological Survey, Vilnius (Karatajūtė-Talimaa & Novitskaya, 1992).

Material

Several hundred isolated scales from the type locality (from samples P-16/3 and ЦГЭ N1009). Non-figured specimens examined for this study are stored in the microvertebrate research collection of the Lapworth Museum of Geology, University of Birmingham, UK.

Description

Morphology

The number of the scale odontocomplex rows is related to crown size and its proportions. In small specimens (less than 0.5 mm long) their number varies from 4 to 6, whilst it reaches 17 in scales larger than 1 mm. Within the individual odontocomplexes the odontode length gradually increases in a posterior direction (Fig. 6F), whereas odontode thickness remains relatively constant at c. 50 µm.

In the majority of specimens a crescent-shaped platform (Figs. 3E and 3F) is formed anterior to the odontocomplexes, and the former can be elevated slightly above the level of the odontodes. The absence of this thickening does not correlate with a particular scale size.

The base is not constricted at the contact with the crown (Fig. 3E–3G) and extends away from this junction into an anteriorly-directed conical projection that protrudes beyond the crown margin. The posterior third of the base is shallower in comparison with its thickened anterior (Fig. 6F), and is marked by rows of canal openings (30–60 µm in diameter; Fig. 3G) aligned with the odontocomplexes of the crown.

Histology

The crown odontodes consist of atubular dentine (lamellin; Fig. 6F) having a predominately lamellar periphery and an inner spheritically mineralised region. The calcospherites of the globular lamellin attain a diameter of approximately 10 µm and comprise of concentric Liesegang rings closed around a central cavity. These exhibit linear or concave arrested growth contact surfaces with other spherites and adjacent Liesegang waves. The scale odontodes possess vascular spaces in the form of vestiges of pulp canals that are mostly filled by lamellin. The pulps branch out laterally as paired short horizontal canals (diameter 10–15 µm) that open on the odontode surface (Fig. 8D, D1).

A structural variety of atubular dentine different from lamellin forms the crown platform that surmounts the thickest part of the base (Fig. 6F). This tissue exhibits exclusively spheritic mineralisation represented by tightly packed globules (up to 10 µm in diameter), and lacks a canal system.

The basal bone is acellular with a series of depositional lamellae demarcated by basally arched intrinsic fibres (Fig. 6F). The smallest lamellae reside at the level of the anterior-most odontodes, with lamella thickness varying from 15 µm to 20 µm across the extent of the tissue.

The basal bone contains extrinsic mineralised fibres grouped into 20–40 µm thick layers with upwardly curved profiles. The fibres within each layer are mutually parallel but also oriented obliquely to those of adjacent lamellae, giving the bone a plywood-like texture. In addition to the abundant fibres with layered organization, the tissue contains a set of extrinsic, vertically oriented fibres (Fig. 6F) that are evenly spaced at about 5 µm intervals and propagate up to the level of the crown-base junction.

The base is penetrated by a number of large-calibre vertical vascular canals (Figs. 8D and 8D1), which connect with the pulp cavities of crown odontodes. The former are predominantly preserved in the posterior (thinnest) third of the base as anteriorly arching canals that gradually widen to c. 40 µm at the lower base surface (Figs. 8D and 8D1).

Remarks

The anterior crown platform of Teslepis scales (developed also in Sodolepis) received little attention in the descriptions of Karatajūtė-Talimaa & Novitskaya (1992) and Karatajūtė-Talimaa (1998), apart from being identified as composed of an undetermined type of globular basal tissue. The platform always forms at the level of the primary odontodes and sutures to the anterior most of them, developing in the space typically occupied by secondary odontodes in Mongolepis, Rongolepis, Xinjiangichthys and Shiqianolepis scales. From a histological perspective, the lack of lamellar matrix and the predominantly arrested-growth contact surfaces of spherites resemble the microstructure of certain types of spheritically mineralized dentine (Schmidt & Keil, 1971, Figs. 46 and 47). Consequently, this tissue is regarded to be globular atubular dentine as opposed to globular dermal bone that is commonly formed only in the cavity-rich cancellous zone of the exoskeleton of lower vertebrates (Ørvig, 1968; Donoghue, Sansom & Downs, 2006; Downs & Donoghue, 2009).

Contrasting with the well-defined and consistent shape of the odontodes, the anterior platform has an irregular surface and poorly defined boundaries, and whose shape is determined by the contours of the underlying base. As a consequence, it could be suggested that this mass of globular dentine is not the product of a well-differentiated dermal papilla, which typifies early odontode development and determines the morphology of odontodes independently of that of the basal bone (Sire, 1994; Sire & Huysseune, 1996; Sire & Huysseune, 2003). Outside Teslepis and Sodolepis, dentine structures with similar characteristics have not been documented in the integumentary skeleton of gnathostomes.

Cellular basal bone was considered by Karatajūtė-Talimaa & Novitskaya (1992) to be a diagnostic characteristic of Teslepis in the original description of the genus. Fusiform odontocyte lacunae identified in that study are considered herein to represent hollow interiors of mineralised fibres of within bone matrix (the implications of this revised interpretation are expanded on in the Discussion).

Genus SODOLEPIS Karatajūtė-Talimaa & Novitskaya, 1997

Type and only species

Sodolepis lucens Karatajūtė-Talimaa & Novitskaya, 1997, from the Chargat Formation (Salhit regional Stage, Upper Llandovery–Lower Wenlock) of north-western Mongolia.

Diagnosis

As for the type species.

SODOLEPIS LUCENS Karatajūtė-Talimaa & Novitskaya, 1997 (Figs. 3H–3J, 6G–6J, 8E)

1997 Sodolepis lucens Karatajūtė-Talimaa & Novitskaya, Figs. 1–3, pl. XI.	
1998 Sodolepis lucens Karatajūtė-Talimaa, Fig. 18.	

Emended diagnosis

Mongolepidids with medium-sized scales (up to over 2 mm) possessing crowns composed of sutured odontocomplex rows, whose number does not increase with scale size. Anterior crown platform of globular dentine elevated to the level of the crown surface. Neck (horizontal) canals not formed at the lower portion of crown odontodes.

Holotype

An isolated scale with accession number LGI M-1-091 deposited in collection LGI M-1 of the Lithuanian Geological Survey, Vilnius (Karatajūtė-Talimaa & Novitskaya, 1997).

Referred material

More than a hundred isolated scales from the type locality (samples P-16/3 and ЦГЭ N1009). Non-figured specimens examined for this study are stored in the Lapworth Museum of Geology, University of Birmingham, UK.

Remarks

The gross morphology of Sodolepis scales (Figs. 3H–3J) closely resembles that of Teslepis, with the two genera demonstrating comparable histology. The latter, however, are distinguished on the basis of differences in odontode size and crown vascularization. Sodolepis crowns possess fused odontocomplexes, composed of odontodes that are on average three times as large of those of Teslepis, divided by inter-odontocomplex spaces. This is due to a corresponding increase of odontode and scale size in Sodolepis, leading to the formation of a relatively constant number of odontocomplexes irrespective of crown dimensions. In Teslepis specimens, on the other hand, odontode size remains consistent across all documented scale lengths.

As noted by Karatajūtė-Talimaa & Novitskaya (1997), a system of horizontal canals cannot be identified inside Sodolepis scale crowns (Fig. 8E)—an atypical condition considering that the majority of mongolepid genera, including Teslepis, develop some type of pulp canal openings on the lower crown surface.

Genus RONGOLEPIS Sansom, Aldridge & Smith, 2000

Type and only species

Rongolepis cosmetica from the Telychian (Upper Llandovery) of south China, Lower Member of the Xiushan Formation (Sansom, Aldridge & Smith, 2000) and the Telychian of Bachu County, Xinjiang, China (Lower member of the Yimugantawu Formation; NZ Wang, 2011, unpublished data).

Diagnosis

As for the type species.

RONGOLEPIS COSMETICA Sansom, Aldridge & Smith, 2000 (Figs. 3K–3M, 6K, 6L)

2000 Rongolepis cosmetica Sansom, Aldridge & Smith, Figs. 11, 12.	

Emended diagnosis

Mongolepidid species with scale odontocomplex rows ornamented by narrow median ridges, flanked anteriorly and laterally by conical secondary odontodes. Posterior primary odontodes long and straight, having pitted by rows of foramina on their lower crown face. Base tetragonal or oblong, displaced towards the scale anterior. Lower base surface concave to flat with a central conical projection.

Holotype

An isolated scale (NIGP 130326) from the Xiushan Formation of south China (Sansom, Aldridge & Smith, 2000).

Referred material

Hundreds of specimens from the Xiushan Formation of Leijiatun (Shiqian county, south China (sample Shiqian 14B), including type series material (NIGP 130319–NIGP 130330) figured by Sansom, Aldridge & Smith (2000). Non-figured specimens stored in the Nanjing Institute of Geology and Palaeontology, Chinese Academy of Sciences, Nanjing, China.

Remarks

The uncertainty regarding the supergeneric position of Rongolepis in the original description of the genus (Sansom, Aldridge & Smith, 2000) has been attributed to a suite of characteristics (scale morphology, posterior of the crown composed of acellular lamellar bone and presence of crown odontodes) not known in the scales of other vertebrates. The re-examination of Rongolepis cosmetica has enabled the identification of a combination of features diagnostic for Mongolepidida. Of particular importance in this regard is the nature of the tissue composing the flared posterior extension of Rongolepis scales. Suggested to be formed of lamellar bone (Sansom, Aldridge & Smith, 2000), this portion of the scale in fact demonstrates the lamellin-type architecture of an inotropically and spheritically mineralized (for definitions of both see Ørvig, 1968; Zylberberg et al., 1992) atubular tissue devoid of attachment fibres (Figs. 6K and 6L). Moreover, the segmentation of the crown’s posterior part observed in thin sections (Figs. 6K and 6L; Sansom, Aldridge & Smith, 2000, Fig. 12E) is interpreted to be produced by the contact surfaces of sutured odontodes. Both the anterior to posterior increase in length of these elements and their arrangement in longitudinal rows over the posterior half of the base are known features of mongolepid primary odontocomplexes. The assignment of Rongolepis to Mongolepidida is thus dictated by the possession of its scales of lamellin and polyodontocomplex growing crowns.

Family SHIQIANOLEPIDAE Sansom, Aldridge & Smith, 2000

Included genera

Xinjiangichthys Wang et al., 1998 and Shiqianolepis Sansom, Aldridge & Smith, 2000.

Emended diagnosis

Mongolepids with scale bases composed of non-vascular, cellular bone tissue.

Genus SHIQIANOLEPIS Sansom, Aldridge & Smith, 2000

Type and only species

Shiqianolepis hollandi Sansom, Aldridge & Smith, 2000, from the Telychian Lower Member of the Xiushan Formation (Leijiatun, Shiqian county, southern China).

Emended diagnosis

As for the type species.

SHIQIANOLEPIS HOLLANDI Sansom, Aldridge & Smith, 2000 (Figs. 4A–4C, 5N, 8F, 9B, 9E)

2000 Shiqianolepis hollandi Sansom, Aldridge & Smith, Figs. 4–6.	

Emended diagnosis

Shiqianolepids with trunk scale odontocomplexes separated posteriorly by deep inter-odontocomplex spaces. A cluster of tightly sutured secondary odontodes formed anteriorly of crown odontocomplexes. Crown surface ornamented by tuberculate ridges. Oblong asymmetrical head scales (up to 1 mm long) with irregularly-shaped odontodes distributed peripherally around a medial ridge.

Holotype

An isolated trunk scale (NIGP 130294) from the Xiushan Formation of Leijiatun (Shiqian County) south China (Sansom, Aldridge & Smith, 2000).

Referred material

Hundreds of isolated scales and type series specimens (NIGP 130293–NIGP 130318) figured by Sansom, Aldridge & Smith (2000) from the Telychian Xiushan Formation (sample Shiqian 14B) of Leijiatun (Shiqian county, south China). Non-figured material stored in the Nanjing Institute of Geology and Palaeontology, Chinese Academy of Sciences, Nanjing, China.

Remarks

Characteristic for Shiqianolepis scales is a distinct primordial odontode located at the apex of the conical base. This odontode has been termed ‘proto-scale’ by Sansom, Aldridge & Smith (2000) and was identified as a diminutive element overlain by the much larger odontodes deposited at later stages of crown ontogeny. Superpositional growth, which results in odontodes not being exposed on the crown surface, is a condition atypical for other mongolepids, also demonstrated to not be a feature of Shiqianolepis scales. Upon re-examination of figured material and newly sectioned specimens, the primordial odontode borders recognized in Sansom, Aldridge & Smith (2000, Figs. 6B and 7) are now considered to constitute the margins of dentine depositional lamellae (Fig. 6N), as these are occasionally observed to be indented by more peripherally formed calcospherites—evidencing a centripetal mode of dentine histogenesis as opposed to stacking of primary odontodes. As identified here, the primordial odontode in Shiqianolepis scales is overlapped only at its anterior end by secondary odontodes, whilst most of its upper margin remains exposed on the crown surface. Similarly to the rest of the odontocomplexes of Shiqianolepis trunk scales, the one incepted by the ‘proto-scale’ also displays a gradual posterior increase of odontode size.

Genus XINJIANGICHTHYS Wang et al., 1998

Type and only species

Xinjiangichthys pluridentatus Wang et al., 1998, from the Telychian Yimugantawu Formation (north-western margin of the Tarim Basin, Xinjiang, PR China).

Emended diagnosis

As for the type species.

Remarks

The placement of Xinjiangichthys inside Mongolepidida by Wang et al. (1998) was justified on the grounds of similarities in crown morphology and odontode patterning with Mongolian mongolepids (the only known mongolepid taxa at the time of its description), and this study advances that claim further by identifying a polyodontocomplex crown structure in Xinjiangichthys scales.

The presence of atubular dentine in Xinjiangichthys scales, another of the diagnostic characters of mongolepids (this study; Karatajūtė-Talimaa et al., 1990; Sansom, Aldridge & Smith, 2000), can be determined in thin-section (Fig. 6M) and through X-ray microtomography (Figs. 8G and 8H).

Furthermore, Wang et al.’s (1998) interpretation of Xinjiangichthys scale bases as non-growing is rejected here by the recognition of a conical basal tissue that supports, at its apex, the primordial odontode and further posteriorly the rest of the scale’s primary odontodes, similarly to the growing bases of Shiqianolepis and those of mongolepids in large (Figs. 6M and 8H).

XINJIANGICHTHYS PLURIDENTATUS Wang et al., 1998 (Figs. 4D–4F, 6M, 8G–8H)

1998 Xinjiangichthys pluridentatusWang, Zhang, Wang and Zhu, pl. 1, Fig. A–D.	
1998 Xinjiangichthys tarimensis Wang, Zhang, Wang & Zhu, pl. 1, Fig. E–I.	
v. 2000 Xinjiangichthys sp. Sansom, Aldridge and Smith, 236, Fig. 8.	

Emended diagnosis

Shiqianolepids with unornamented scale crowns composed of sutured odontocomplex rows. Needle-like primary odontodes; erect, conical secondary odontodes.

Holotype

An isolated trunk scale (IVPP V11663.1) from the Yimugantawu Formation of Xinjiang (Bachu county), China (Wang et al., 1998).

Referred material

Two specimens from the Telychian Xiushan Formation (Leijiatun, Shiqian county, south China; sample Shiqian 14B), in addition to material figured (NIGP 130291, NIGP 130292) in Sansom, Aldridge & Smith (2000), and five specimens (including IVPP V X1, IVPP V X2) from the Yimugantawu Formation (Bachu county, Xinjiang, PR China). Non-figured scales are stored in the Nanjing Institute of Geology and Palaeontology, Chinese Academy of Sciences, Nanjing, China and the Institute of Vertebrate Paleontology and Paleoanthropology, Chinese Academy of Sciences, Beijing, China.

Remarks

X. tarimensis and X. sp. are synonymised with X. pluridentatus based on the absence of differentiating characteristics between the specimens attributed to the two species. The arguments (equal-sized crown odontodes, scale neck and pitted sub-crown surface) of Wang et al. (1998) for erecting X. tarimensis are considered not valid for the following reasons. The large-diameter anterior odontodes of X. pluridentatus specimens figured by Wang et al. (1998, pl. Ia, c) represent secondary odontodes not developed in all scales of the species (specimens identified as X. tarimensis by Wang et al., 1998, pl. Ie-i), which is consistent with the condition documented in Mongolepis (this study and Karatajūtė-Talimaa et al., 1990). The presence of secondary odontodes also accounts for the lack of a distinct neck in the Xinjiangichthys scales they develop, by occupying the sloped anterior surface of the base. The third character considered diagnostic for X. tarimensis by Wang et al. (1998) are the numerous foramina present on the lower crown surface of scales, which are also seen (Figs. 4D, 4E and 8G–8H) in Xinjiangichthys specimens with secondary odontodes.

Family incertae sedis

Genus SOLINALEPIS gen. nov.

Type and only species

Solinalepis levis gen. et sp. nov.

Derivation of name

From ‘solinas’ (tube, pipe in Greek), pertaining to the shape of the scale odontodes of the species, and ‘lepis’, scale in Greek.

Diagnosis

As for the type species.

Remarks

Characters relating to the dimensions of the scale base (its extent and thickness in relation to those of the crown) unite Solinalepis gen. nov. (data from yet to be published phylogenetic analysis by Andreev et al., in press; Fig. 2) in a clade with members of Shiqianolepidae. Nevertheless, this type of morphological data is not regarded informative at a supra-generic level and the genus is classified outside the two recognized mongolepid families due to differences in scale base histology (acellular bone lacking plywood-like organization of its mineralised matrix). As a consequence, Solinalepis gen. nov. is treated as Mongolepidida incertae sedis.

SOLINALEPIS LEVIS sp. nov (Figs. 5, 7, 8I–8J, 9C)

Figure 8 Canal system of mongolepid scales.

Volume renderings. (A–C) canals (red) inside a translucent Mongolepis rozmanae scale (BU5296) in (A) lateral view, in (B) posterior view sliced along the plane 1 and in (C, C1) crown view sliced along plane 2; (D, D1) canals in a transversely sliced Teslepis jucunda scale (BU5325) shown in posterior view; (E) pulp cavities (red) in a transversely sliced Sodolepis lucens scale (BU5305) shown in postero-lateral view; (F) longitudinally sliced Shiqianolepis hollandi scale (NIGP 130307) in baso-lateral view; (G, H) longitudinally sliced Xinjiangichthys pluridentatus scale IVPP V X2 in (G) posterior and (H) lateral views; (I, J) canals system (red) inside a transversely sliced Solinalepis levis gen. et sp. nov. scale (BU5318) shown in posterior view, (J) detail of (I). Horizontal canals depicted in purple in c1 and d1. Yellow arrowheads point at canal openings on the sub-crown surface. Red dotted line, contact surfaces between primary and secondary odontodes; grey dotted line, crown/base border. Scale bar equals 400 µm in (A–C), 100 µm in (D, H, I), 200 µm in (E), 300 µm (F, G) and 50 µm in (J).

Figure 9 Odontocomplex organization of mongolepid scale crowns.

(A) Teslepis jucunda (BU5323) scale, medial portion of the crown; (B) Shiqianolepis hollandi (NIGP 130309) scale, medial portion of the crown; (C) Solinalepis levis gen. et sp. nov. trunk scale (BU5314), lateral portion of the crown. Primary odontocomplex structure in Mongolepidida demonstrated by line drawings of longitudinally sectioned (D) Mongolepis rozmanae (BU5297) and (E) Shiqianolepis hollandi (NIGP 130312) scales. In (A–C) some of the odontocomplexes are highlighted in red and green. Dark green and dark red, odd numbered odontodes; light green and light red, even numbered odontodes. In (D, E)—light grey, primary odontodes; light yellow, secondary odontodes. Anterior towards the bottom in (A–C) and towards the left in (D, E). Scale bar equals 100 µm in (A), 200 µm in (B) and 50 µm in (C).

2001 ‘?Mongolepid scales’ Sansom, Smith and Smith, p. 161, Figs. 10.3G, 10.3H.	
2002 Unnamed chondrichthyan Donoghue and Sansom, p. 362, Fig. 6.3.	
2009 Stem-chondrichthyan Sire, Donoghue and Vickaryous, p. 424, Fig. 10C.	

Derivation of name

From the Latin ‘levis’ (smooth), referring to the unornamented scale crown surface of the species.

Locality and horizon

The type locality is the vicinity of the Harding Quarry, situated c. 1 km west of Cañon City (Fremont County, Colorado, USA). All Solinalepis specimens come from Sandbian strata (Mohawkian regional series, Phragmodus undatus conodont zone) of the Harding Sandstone (samples H94-26 and H96-20).

Holotype

An isolated trunk scale BU5310 (Fig. 5E).

Referred material

Hundreds of isolated scales, including BU5307–BU5318, BU5345. Non-figured specimens examined for this study are stored in the microvertebrate research collection of the Lapworth Museum of Geology, University of Birmingham, UK.

Diagnosis

Mongolepid species with trunk scales crowns composed of tubular odontodes organized in sutured longitudinal odontocomplex rows. Acellular basal bone housing an elaborate canal system that opens via foramina on the basal surface. Radially arranged tuberculate to conical head-scale odontodes.

Description

Morphology of head scales

Polyodontode symmetrical or asymmetrical scales with height between 0.5 and 1.3 mm. These are represented by two main morphological variants, a compact, bulbous type (Fig. 5D) and tessera-like scales (Figs. 5A–5C) of larger diameter. Both morphotypes possess irregular crowns composed of radially ordered odontodes, and do not clearly exhibit distinct anterior, posterior and lateral scale faces. The radiating odontodes form rows (five to nine odontodes long), offset in a manner in which the odontodes of each row oppose the inter-odontode contacts of neighbouring odontocomplexes. Odontode height diminishes gradually towards the crown centre, accompanied by an increase of coalescence between odontodes.

The scales exhibit a prominent central bulge, away from which the crown surface slopes down to the scale margin. In crown view, the latter has a corrugated outline that in certain specimens is accentuated by deep, peripherally expanding grooves (Figs. 5A and 5B).

The scale base displays a granular, grooved surface and follows the outline of the crown. At its centre the base attains maximal thickness (Fig. 7A), and gradually decreases in height away from this point. The lower-base surface is predominantly planar or can have a moderate central concavity, but never exhibits the convex topology documented in trunk scale specimens.

Morphology of trunk scales

The length of these scales varies between 100–400 µm and is always less (up to three quarters) than their width. Specimens with crown lengths near or exceeding 200 µm demonstrate polygonal (Figs. 5E–5G), often asymmetrical (Figs. 5F and 5G), outlines. The anterior crown margin of these scales is typically wedge-shaped whilst their posterior face is straight (Fig. 5I). In contrast, the crowns of antero-posteriorly short (100–200 µm long) scales tend to be symmetrical, leaf-shaped structures (Figs. 5J–5L), rarely demonstrating simple geometrical profiles in crown view.

Irrespective of crown morphology, the odontodes of trunk scales are organized into closely packed antero-posteriorly aligned rows (Figs. 5F–5G, 5J and 9C). Adjacent rows are displaced by approximately half an odontode diameter (c. 15 µm), resulting in an offset between the odontodes of neighbouring odontocomplexes (Fig. 9C). The odontodes themselves are cylindrical, tube-like elements with sigmoidal profiles that taper to a point apically (Fig. 5J). Odontode length increases gradually towards the scale’s posterior end, where the crown can reach a height of c. 400 µm.

The crown/base transition is not marked by a neck-like constriction (Figs. 5E–5L), with the base never attaining more than a third of the overall scale height. The basal surface is typically marked by deeply incised grooves (Figs. 5E–5I) that give it a dimpled appearance, characteristic also for the lower base surface. The latter has a predominantly flat profile but can exhibit a central conical projection that is particularly well developed in leaf-shaped specimens (Fig. 5L).

Histology of head scales

Due to diagenetic alteration of histologically examined scales, the microstructure of crown odontodes is largely obscured. Nevertheless, wide odontode pulp canals are evident in sectioned specimens (Fig. 7A), and these appear to end blindly inside the crown. The upper base surface is perforated by a row of foramina (Figs. 5C and 5D) similar to the ones documented in trunk scales.

The main structural components of the basal bone matrix are tightly packed, parallel crystalline mineralized fibres with horizontal orientation (Fig. 7A). These are crosscut by apically converging fibre bundles (up to 15 µm in diameter), which follow undulating paths across the tissue.

Histology of trunk scales

Crown odontodes are structured out of atubular dentine (lamellin; Fig. 7B) that is spherically mineralised in proximity of the pulp (spherite diameter 10–15 µm).

Cylindrical, non-branching pulp cavities occupy the centre of odontodes and are connected at their lower ends with the canal system of the base (Figs. 8I and 8J). The latter is represented by vertical canals that bifurcate close to the crown-base junction, with each pair of rami re-connecting deeper inside the base, resulting in the formation of a series of vascular loops (Figs. 8I and 8J). Vertically oriented canals emerge from the looped canal system and open on the lower base surface. The basal surface is similarly marked by numerous foramina that are the exit points for the peripheral canals of the base (Fig. 5H).

The base is composed of acellular bone demonstrating the presence of c. 2 µm thick extrinsic crystalline mineralised fibres that propagate vertically through the tissue (Fig. 7B).

Remarks

The development of polyodontocomplex scale crowns formed from lamellin identify Solinalepis levis gen. et sp. nov. scales as a mongolepid species. Moreover, the trunk scale odontocomplexes of Solinalepis gen. nov. exhibit the same progressive posterior increase in odontode length documented in members of the Order.

Within Mongolepidida, the combination of a large odontocomplex number (>20) and sutured odontodes is present only in the Telychian genus Xinjiangichthys. Nevertheless, the two taxa are readily distinguished on the basis of base histology and canal-opening distribution on the scale surface. In addition to that, Solinalepis gen. nov. is one of only two described mongolepid genera (the other being Shiqianolepis) known to develop with squamation clearly differentiated into distinct trunk (exhibiting recognizable anterior and posterior faces) and head morphotypes (irregular-shaped elements)—a condition that is consistent with that recorded in a number of heterosquamous Lower Palaeozoic gnathostomes known from articulated specimens (e.g., Climatius reticulatus Miles, 1973, Obtusacanthus corroconius Hanke & Wilson, 2004, Gladiobranchus probaton Hanke & Davis, 2008 and Ptomacanthus anglicus Miles, 1973; Brazeau, 2012).

Discussion

Crown morphogenesis of mongolepid scales

Shiqianolepis hollandi is recognized as a key taxon for determining the mode of scale crown development in mongolepids, following the identification by Sansom, Aldridge & Smith (2000) of ‘proto-scale’ (early-development phase) specimens of the species (Sansom, Aldridge & Smith, 2000, Figs. 4U and 4W). The size (half of that of ‘mature’ trunk scales) and the small number of crown odontodes (exhibiting only the earliest formed odontodes of incipient primary odontocomplexes) of these scales implies that in Shiqianolepis scale ontogenesis involves crown enlargement through sequential addition of odontodes. Significantly, this style of crown architecture (primary odontocomplex rows originating at the most elevated point of the base and characterized by a posterior increase in size of their constituent odontodes) is developed in all members of the Mongolepidida (Figs. 6A, 6F, 6I, 6K, 6M, 6N and 9) and is evidence that the mongolepids share a cyclomorial pattern of scale ontogenesis.

Data from developmental studies on extant neoselachians indicate that their scales cannot serve as model systems for determining the mechanism of morphogenesis of the compound mongolepid scale crowns, as the former have been shown to be simple monodontode elements produced by a single epithelio-ectomesenchymal primordium (Schmidt & Keil, 1971; Reif, 1980; Miyake et al., 1999; Sire & Huysseune, 2003; Johanson, Smith & Joss, 2007; Johanson et al., 2008). Examinations of multiple odontode generation in osteichthyan scales (Kerr, 1952; Smith, Hobdell & Miller, 1972; Smith, 1979; Sire & Huysseune, 1996), though, provide insight into the timing of deposition of odontode aggregations associated with a dermal bone support tissue. They reveal phases of odontode generation that result in an increase of odontode number throughout scale ontogeny.

The proposed scale growth mechanism in Mongolepidida is further substantiated by evidence from the Palaeozoic record of the Chondrichthyes. The scale crown structure of certain euchondrichthyan taxa described from articulated specimens (e.g., Diplodoselache woodi Dick, 1981, Tamiobatis vetustus Williams, 1998 and Orodus greggi Zangerl, 1968), conform closely to the odontode patterning of mongolepid scales. Diplodeselache trunk scales were noted by Dick (1981) to closely resemble those of Orodus and to be similarly characterized by cyclomorial growth. Previous work (Reif, 1978) on the morphogenesis of the chondrichthyan integumentary skeleton also recognized sequential crown elongation through regular addition of odontodes as the mechanism of scale development in Orodus. This pattern of crown formation is also typical for scales with a Ctenacanthus costellatus type of morphogenesis (defined by Reif, 1978 and equivalent to the Ctenacanthus B3 morphogenetic type of Karatajūtė-Talimaa, 1992) to which Tamiobatis scales have been attributed (Williams, 1998).

Mongolepid scale crown histology

The emergence of skeletal mineralisation in vertebrates (Donoghue & Sansom, 2002; Donoghue, Sansom & Downs, 2006) coincides with the origin of atubular dentine-like tissues that compose the basal bodies of certain conodont genera (Sansom, 1996; Smith, Sansom & Smith, 1996; Donoghue, 1998; Dong, Donoghue & Repetski, 2005). Conodont atubular ‘dentines’ frequently exhibit (Sansom, 1996, Figs. 2E–H; Donoghue, 1998, Figs. 5A–C; Dong, Donoghue & Repetski, 2005, pl. 1, Figs. 3–9) peripheral lamellar fabric, substituted internally by spheritically mineralised matrix, making them structurally (but not phylogenetically) comparable with the architecture of mongolepid lamellin (Figs. 6C and 6G). The conodont tissues have recently been hypothesized to have arisen in a stepwise manner in the oropharyngeal skeleton of the Paraconodonta and Euconodonta (Murdock et al., 2013), whilst separately, within the Total Group Gnathostomata the known occurrence of atubular dentines outside the Mongolepidida is limited to the scale odontodes of the pteraspidomorph Tesakoviaspis concentrica (Karatajūtė-Talimaa & Smith, 2004) and the fin spine ornament of sinacanthid gnathostomes (Sansom, Aldridge & Smith, 2000; Sansom, Wang & Smith, 2005).

An important aspect of the atubular nature of lamellin is that it provides circumstantial evidence for the involvement of atypical (from a modern perspective) odontoblasts in the generation of the tissue. During dentinogenesis mature odontoblasts commonly extend long cellular processes into the mineralised phase, which remain contained inside tubular spaces after formation of the tissue is complete (Linde, 1989; Linde & Lundgren, 1995; Yoshiba et al., 2002; Magloire et al., 2004; Magloire et al., 2009). The inability of secretory odontoblasts to form dentinal tubules is taken to suggest that such cells either did not embed their processes within the dentine matrix at any depth or lacked processes altogether. Atypical odontoblasts devoid of large cytoplasmic projections have been reported in the tooth germs of the Recent sting ray Dasyatis akajei (Sasagawa, 1995), but these are found to co-exist with unipolar odontoblasts, characterized by well-developed processes. The apical portions of odontoblasts and their processes have been implicated as ion channel-rich sites capable of being activated by environmental stimuli via tubular fluid movement, and are presumably involved in transmitting sensory input to pulp nerve endings (Okumura et al., 2005; Allard et al., 2006; Magloire et al., 2009). This raises the possibility that mongolepid scale pulps had limited ability to transduce sensory input compared with an odontoblast population that forms tubular network inside a mineralised dentine matrix.

Histology of mongolepid scale bases

This and previous studies (Karatajūtė-Talimaa et al., 1990; Karatajūtė-Talimaa, 1995; Karatajūtė-Talimaa & Novitskaya, 1992; Karatajūtė-Talimaa & Novitskaya, 1997; Sansom, Aldridge & Smith, 2000) identify mongolepid scale odontodes to be supported by a common base composed of lamellar bone (Figs. 6A, 6F, 6H, 6I, 6K, 6M, 6N and 7). The basal tissue of Mongolepis and Sodolepis scales has been interpreted as acellular bone (Karatajūtė-Talimaa et al., 1990; Karatajūtė-Talimaa & Novitskaya, 1997), with this study also recognizing the absence of osteocyte lacunae in the bases of Teslepis (contra Karatajūtė-Talimaa & Novitskaya, 1992), Rongolepis (in agreement with Sansom, Aldridge & Smith, 2000) and Solinalepis gen. nov.—restricting the occurrence of cellular bone inside Mongolepidida to the genera Xinjiangichthys and Shiqianolepis (this study and Sansom, Aldridge & Smith, 2000).

A cross-ply layering of crystalline fibres is recognized as the predominant type of basal bone texture of mongolepid scales, being documented in the four genera of the Family Mongolepididae. This architecture of the mineralised matrix matches closely the organization of the collagen fibres in the deep dermis (stratum compactum) of extant neoselachians (Motta, 1977; Miyake et al., 1999; Sire & Huysseune, 2003) and osteichthyans (Kerr, 1952; Kerr, 1955; Sire, 1993; Gemballa & Bartsch, 2002) and is suggested to be indicative of dermal bone histogenesis achieved through mineralisation of the a largely unmodified fibrous scaffold of the stratum compactum—a process referred to as metaplastic ossification (Sire, 1993; Sire, Donoghue & Vickaryous, 2009). Consequently, the observed absence of cross-ply layering in the cellular bone of mongolepid scale bases (in Xinjiangichthys, Shiqianolepis and Solinalepis gen. nov.) could be interpreted to result from remodeling of the original fibrous framework of stratum compactum prior to tissue mineralisation (a process described by Sire (1993) in the scales of the armoured catfish Corydoras arcuatus).

The data above allow the identification of the site of basal bone formation of mongolepid scales within the deep tiers of the corium, with the tissue being considered to periodically increase in size due to the growth increments documented in sectioned specimens. These depositional phases reveal a common pattern of generation of mongolepid scale bases, wherein each newly laid down lamella covers the lower surface of the previously deposited one. The geometry of the lamellae shows little change, implying retention of a fairly consistent base shape throughout scale ontogeny. Such a pattern of base morphogenesis is not unique to the Mongolepidida, but appears to be the prevalent mode of bone tissue growth in the scales of jawed gnathostomes, being demonstrated in ‘placoderms’ (Burrow & Turner, 1998; Burrow & Turner, 1999), ‘acanthodians’ (Denison, 1979), early osteichthyans (Gross, 1968; Schultze, 1968) and early chondrichthyans (Karatajūtė-Talimaa, 1973; Mader, 1986; Wang, 1993).

Canal system of mongolepid scales

Previously, the internal canal system architecture of mongolepid scales had been investigated in detail only in Mongolepis, Teslepis and Sodolepis through oil immersion studies and thin section work (Karatajūtė-Talimaa et al., 1990; Karatajūtė-Talimaa & Novitskaya, 1992; Karatajūtė-Talimaa & Novitskaya, 1997). The employment of X-ray microtomography extended to these observations by enabling visualization of the three-dimensional structure of scale cavity spaces in the examined genera with greater accuracy.

In Mongolepis, Teslepis, Sodolepis and Solinalepis gen. nov. the lower ends of odontode pulp cavities are continuous with the canal system of the base. Comparable vascularization is developed in the Upper Ordovician chondrichthyan scale species Tezakia hardingensis from North America (Andreev et al., 2015). The lower base surface of this taxon has been demonstrated to exhibit rows of foramina (Sansom, Smith & Smith, 1996, Fig. 2A) that are similar to the basal canal openings of mongolepids. Likewise, the central canal of the basal bone tissue is continuous with the odontode pulp in the Silurian scale genera Elegestolepis (Karatajūtė-Talimaa, 1973; Andreev et al., in press) and Kannathalepis (Märss & Gagnier, 2001), which are the earliest recorded monodontode scale taxa attributed to the Chondrichthyes (Andreev et al., in press). This condition is also identified in the monodontode scales of various Upper Palaeozoic chondrichthyans (e.g., Janassa Ørvig, 1966; Malzahn, 1968, Ornithoprion Zangerl, 1966 and Hopleacanthus Schaumberg, 1982), Mesozoic hybodonts (Reif, 1978) and extant neoselachians (Reif, 1980; Miyake et al., 1999; Johanson et al., 2008).

Xinjiangichthys, Shiqianolepis and Rongolepis differ from the other mongolepid genera in having their entire scale canal system confined to the crown, with the lower ends of odontode pulps opening at the crown surface in proximity of the base. The posterior peripheral odontodes of these three genera display additional cavities that are detected as foramina on the lower crown face. A similarly pitted lower crown surface has also been identified in poracanthodid ‘acanthodians’ (Gross, 1956; Valiukevičius, 1992; Burrow, 2003), the putative stem chondrichthyan Seretolepis (Hanke & Wilson, 2010; Martínez-Pèrez et al., 2010), and in ctenacanthiform scales (e.g., Tamiobatis vetustus Williams, 1998 and Ctenacanthus costellatus Reif, 1978). In the scales of Poracanthodes these openings represent the posterior exit points of a complex canal network that is absent from mongolepid scale crowns.

Studies on the squamation of jawed gnathostomes reveal the lack of basal tissue vascularisation to be a common feature of many ‘acanthodians’ (Denison, 1979; Karatajūtė-Talimaa & Smith, 2003; Valiukevičius, 2003; Valiukevičius & Burrow, 2005) and euchondrichthyans such as Protacrodus (Gross, 1973), Orodus (Zangerl, 1968) and Holmesella (Ørvig, 1966), including some of the earliest known post-Silurian putative chondrichthyan taxa (e.g., Iberolepis and Lunalepis Mader, 1986; Nogueralepis Wang, 1993; Gladbachus Burrow & Turner, 2013).

Despite the observed differences in canal architecture, all mongolepid genera with the exception of Sodolepis develop canal openings exposed on the scale surface in the region the crown-base interface. These foramina represent the termini of canals that are positionally equivalent to, and likely homologues of, the neck canals of euselachians (sensu Reif, 1978). In Mongolepis and Teslepis this connection is established via one pair of short canals (the ‘horizontal canals’ of Karatajūtė-Talimaa et al., 1990; Karatajūtė-Talimaa & Novitskaya, 1992; Karatajūtė-Talimaa, 1998) that emerge from the lower end of each pulp. The data presented here indicate that the horizontal canal system of these two genera is housed inside the scale crown, contrary to previous depictions of the feature at the crown-base junction (Karatajūtė-Talimaa, 1995; Karatajūtė-Talimaa, 1998). In contrast, the lower ends of odontode pulp canals of North American and Chinese mongolepids do not branch out, and either continue inside the base without being exposed on the crown surface (Solinalepis gen. nov.) or open directly onto it (Shiqianolepis and Rongolepis). These features point to notable variation in the vascularization of mongolepid species, which are otherwise remarkably consistent in the development of their scales. However, it is unclear if these differences had any influence on the rates of growth and regeneration of the integumentary skeleton.

Systematic position of the Mongolepidida

Recent phylogenetic investigations of Palaeozoic gnathostomes use only a small subset of generalized scale characters (Brazeau, 2009; Davis, Finarelli & Coates, 2012; Zhu et al., 2013; Giles, Friedman & Brazeau, 2015), and this is likewise true for tree reconstructions of the total group Chondrichthyes, where scale data tend to be minor components of employed character matrices (Lund & Grogan, 1997; Coates & Sequeira, 2001; Grogan & Lund, 2008; Grogan, Lund & Greenfest-Allen, 2012). Chondrichthyan clades instead have often been erected upon tooth characters (Zangerl, 1981; Stahl, 1999; Ginter, Hampe & Duffin, 2010), leaving the position of lower Palaeozoic shark-like scale taxa still unresolved in phylogenetic hypotheses for the Chondrichthyes.

The coherence of the Mongolepidida is reaffirmed here on the basis of an amended character set, which diagnoses the Order by the unique combination of scale growth, polyodontocomplex scale crowns and development of lamellin. The placement of mongolepids within Chondrichthyes, on the other hand, has been questioned in the past on the basis of their atubular dentine (lamellin) crowns and the presence of a horizontal canal system (Karatajūtė-Talimaa & Novitskaya, 1992). This study suggests that the horizontal canals of Mongolepis and Teslepis are equivalent to euselachian neck canals, whilst revealing similar canal spaces in the crown odontodes of Chinese mongolepids. However, neck canals are likewise also known in the scales of ‘placoderms’ (Burrow & Turner, 1998) and basal Palaeozoic osteichthyans (Gross, 1953; Gross, 1968), and might not by a chondrichthyan apomorphy. Also, scale dentine histology appears to vary greatly within the total group Chondrichthyes (e.g., distinct dentine types are developed in Elegestolepis Karatajūtė-Talimaa, 1973, Seretolepis Hanke & Wilson, 2010, Orodus Zangerl, 1968 and Hybodus Reif, 1978), which makes it a poor diagnostic character at a supra-ordinal level. By the same token, although atubular dentine occurs in the Mongolepidida, it is also formed in the dermal skeleton of some pteraspidomorph agnathans (Karatajūtė-Talimaa & Smith, 2004) and fin spines that may be of chondrichthyan origin (Zhu, 1998; Sansom, Wang & Smith, 2005), and therefore is uninformative with respect to the relationships of the Order. The systematic affinities of Mongolepidida are determined instead by a unique combination of scale attributes that are shared with other Palaeozoic chondrichthyan lineages. Reference is made here to the Ctenacanthus-type squamation of certain xenacanthiform (Diplodoselache Dick, 1981), orodontiform (Orodus Zangerl, 1968) and cladodontomorph (e.g., Cladolepis Burrow, Turner & Wang, 2000 and Cladoselache Dean, 1909; P Andreev, pers. obs., 2014) chondrichthyans, characterized by the development of symmetrical trunk scales with multiple crown odontocomplexes that lack cancellous bone, enamel and hard tissue resorption.

Conclusions

The present revision of Mongolepidida established the Order as a natural group of early chondrichthyans characterized by polyodontocomplex growing scales with Ctenacanthus-like crown architecture. However, in agreement with Karatajūtė-Talimaa (1992), the scales of mongolepids are recognized to exhibit a distinct, Mongolepis, type of morphogenesis, on account of their lamellin composed crowns.

The description of the mongolepid genus Solinalepis gen. nov. from the Sandbian of North America, pushes back the first appearance of the Mongolepidida by 20 My and places the origin of the Chondrichthyes in the Ordovician. Together with reports of other shark-like scale taxa from the Ordovician (Sansom, Smith & Smith, 1996; Sansom, Smith & Smith, 2001; Sansom et al., 2012) and the Silurian (Karatajūtė-Talimaa, 1973; Karatajūtė-Talimaa & Predtechenskyj, 1995; Sennikov et al., 2015), this lends further support of an early diversification of putative chondrichthyans (proposed by Karatajūtė-Talimaa, 1992) that preceded a major radiation of nektonic faunas (Klug et al., 2010), coincident with the first known appearance of chondrichthyan teeth and articulated skeletal remains in the Lower Devonian.

Solinalepis material was collected from the Harding Sandstone during fieldwork undertaken as part of research conducted by M. Paul Smith (Oxford) and Moya Smith (King’s College, London), and we are grateful to both for discussions on the nature of these specimens over the years, whilst specimens of Shiqianolepis were made available for study by the late Richard J. Aldridge (Leicester). Rachel Sammons and Michael Sandholzer provided technical assistance during SEM and micro-CT imaging of mongolepid scales at the School of Dentistry, University of Birmingham. We are also grateful to Kate Trinajstic, Živilė Žigaitė and one anonymous reviewer for their constructive and helpful comments on the originally submitted manuscript.

The authors would like to highlight the contribution of the deceased N-Z Wang to this study, who initially examined and provided the Chinese mongolepid material described here.

Additional Information and Declarations

Competing Interests

Author Contributions

Data Availability

New Species Registration

The authors declare there are no competing interests.

Plamen Andreev conceived and designed the experiments, performed the experiments, analyzed the data, wrote the paper, prepared figures and/or tables, reviewed drafts of the paper.

Michael I. Coates wrote the paper, reviewed drafts of the paper.

Valentina Karatajūtė-Talimaa reviewed drafts of the paper, provided part of the studied fossil scale material.

Richard M. Shelton and Paul R. Cooper contributed reagents/materials/analysis tools, reviewed drafts of the paper.

Nian-Zhong Wang analyzed the data, provided part of the studied fossil scale material.

Ivan J. Sansom conceived and designed the experiments, performed the experiments, analyzed the data, wrote the paper, reviewed drafts of the paper.

The following information was supplied regarding data availability:

The research in this article did not generate any raw data.

The following information was supplied regarding the registration of a newly described species:

Publication LSID: urn:lsid:zoobank.org:pub:3C24AE11-1F12-4B16-B04D -480CA204CCEA.

Solinalepis levis LSID: http://zoobank.org/NomenclaturalActs/EB898293-231A-4638-8F1F-89F6C1DABBA0.

Solinalepis LSID: http://zoobank.org/NomenclaturalActs/BCD329FC-82E8-4772-A314-5568CF34C9FC.

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
