# Peer review of "The systematics of the Mongolepidida (Chondrichthyes) and the Ordovician origins of the clade"

_PeerJ, doi:10.7717/peerj.1850_

## Round 0.1 · original submission · Minor Revisions

You present a comprehensive and much-needed synthesis of the taxonomy of mongolepids including computed tomography of scales. The manuscript is therefore of great interest and generally in a good state, which is also supported by the positive assessments of the reviewers. However, there are still some minor points which to be addressed before the manuscript can be published. The main points are:

- Definition of terms: It would be good to briefly define “lamellin” and “Liesegang waves” for non-expert readers

- Unpublished data and submitted phylogenetic analyses: it is hard to verify unpublished data and phylogenetic analyses. The authors should therefore refrain from referring to submitted or unpublished data. The authors should at least discuss these in more detail, particularly specifications and results of the phylogenetic analyses (see also comment by reviewer 3) for comparative purposes.

- Registration numbers for all non-figured specimens: Would it be possible to add numbers to the non-figured specimens? This needs to be addressed throughout the paper (see comments by reviewer 2)

- Diagnosis based on the trunk scales: As remarked by reviewer 2, why is the diagnosis based on trunk scales only, when also head scales are available. It would be easier to follow if the rationale behind this would be discussed in greater detail. Furthermore, even more types of scales might be available if asymmetric scales derive from a fin or lateral line/sensory canals (see comments by reviewer 2)

Please also address the following points in addition to the suggestions made by the reviewers:

Line 6: It would be recommended to add a small sentence at least in the acknowledgements to explain the involvement of the deceased and commemorate his contribution
Line 36, 39: To my knowledge it is not necessary to capitalize “Families” in this context
Line 70: please do not add publications (Andreev et al., submitted) which are submitted and not yet in press or published online
Line 75: I don´t think it is necessary to capitalize “Family” in this context
Line 178: It is questionable to discuss this unpublished data in this context without adding at least the specifications and at least some of the results of the phylogenetic analysis within the current manuscript
Line 217: Could you add details on the numbers of the material stored in the microvertebrate research collection of the Lapwoth Museum
Line 249: “Liesegang waves” might be a common term for experts, but not for the general reader, so please define it and/or at least add a reference which defines it in detail
Line 295: I would replace “refer to” by “see”
Line 446: “N-Z Wang, unpublished”: it might be hard to verify unpublished data by a deceased person
Line 610: I don´t think it is necessary to capitalize “Families” in this context
Line 747: I guess this should be “the herein proposed scale growth mechanism”
Line 880: I find the use of the word “issue” a bit confusing in this context
Line 928: Do you mean “early” or “stem-group” chondrichthyans?
Line 937-939: Interesting hypothesis, but what taphonomic scenarios could have resulted in these discrepancies in the fossil record of scales, teeth and articulated skeletal remains?

·

Basic reporting

No comments

Experimental design

Very good, sophisticated combination of analytical techniques and material involved.

Validity of the findings

No comments

Additional comments

Very well written paper, in-depth study and quality interpretation of the results, excellent illustrations. I do certainly recommend this manuscript to be published, after very minor but necessary corrections: (1) a few related and important citations are missing in the introductory part and previous research overview (see comments in the text); (2) collection numbers of the holotypes from the Lithuanian Institute of Geology (adding 'LGI', see comments in the text and Zigaite 2013 for numbering of the same collection). In the introductory part, I would suggest to include few lines on the 'lamellin', as it is one of the main diagnostic features and is highlighted in the abstract.

·

Basic reporting

The instructions to authors indicate that the corresponding authors are required to provide a street address this didn’t seem to be present.

The figures are referred to a little out of order. I understand that it is better to keep the histology and SEMs of the scales separated and it does make it easier to compare the different scales. However, in plate 4 the trunk scales (4E- L) are discussed before the head scales, but the heard scales are the fist scales figured (4A-D). The first head scales referred to is Fig 4D. This occurs with other figures as well

Does Ordovician need to be repeated in the keywords as it appears in the title?

Experimental design

No comments

Validity of the findings

No Comments

Additional comments

This is a very interesting paper and provides important information on early chondrichthyan taxonomy, histology and radiations. The figures are well presented and additional information has been provided by micro-CT. I certainly recommend this work for publication.

Comments
There is reference to unpublished data from the Tarim Basin within the previous work section. Are these specimens the ones described within this paper? If so then maybe a note can be added. If not then the location and registration of these unpublished specimens may be of use.

Line 78 There is reference to Andrev et al. unpublished phylogenetic data which resolved two monophyletic groups and these characters are used in this work over previously used published characters. As the data not published I cannot evaluate this section

Line 205 Large scales – could a size range or above a certain size be included

Line 215 There are unreadable characters in the last ?word/numeral of the line

Line 216 Non figured material is listed as been housed in the Lapworth Museum Do these specimens have registration numbers and if so could they be listed?

Line 229 "elements primary odontodes" There seems to be a word missing. Should it be "elements of the primary odontodes"

Line 251 pulp canals are either closed off – can a reference to a figure that shows this be provided?

323 The sample notation seems to be lost in the conversion to the PDF thus the last word of the line is ЦГЭ. This will need to be checked throughout.

324 Are there registration numbers for all non-figured specimens? This needs to be addressed throughout the paper.

354 Fig. is missing the period

415 Is there a word missing? Should it be “medium sized scales” and if so can a size range be added?

607 Andreev et al doesn’t have a publication date after it

634 Diagnosis is based only on the trunk scales yet there are head scales described. Why are only the trunk scales used in the diagnosis?

643 Why are the asymmetric scales also considered body scales? Could these represent scales from around a fin or lateral line/sensory canals. The comparison to scale body morphology is largely based on acanthodian body scales, although the taxonomic position of acanthodians as stem sharks is appreciated there are also descriptions of heterosquamous sharks in some of the papers on Palaeozoic sharks that are referred to in the discussion.

676 Figure 6A is referred to in order to demonstrate the base shape and height. Is this correct?

Reviewer 3 ·

Basic reporting

Andreev et al. provide a much-needed and detailed synthesis of the taxonomy of mongolepids—an enigmatic assemblage of scale-based chondrichthyan taxa from the Ordovician and Silurian of Asia and North America. The new data are well presented, well described, and adequately figured. I’m happy to recommend publication. I have attached a few comments to the manuscript.

One issue I would raise is the question of referring to unpublished, unvetted, and unavailable phylogenetic analyses as this manuscript does in a number of places. I would like, ideally, to see some kind of summary cladogram and character optimisations that convincingly show support for the relevant groupings, perhaps something analogous to those used by Brazeau & Friedman (2014. ZJLS). If the authors cannot supply their phylogenetic analysis yet, it would be good to know what it does say in places where these facts are relevant and can be judged by independent parties.

Experimental design

Taxa are appropriately selected and the authors use a relatively novel application of CT to investigate scale morphology for comparative purposes.

Validity of the findings

Overall, the findings appear to be valid, but as detailed above some additional transparency is required for the phylogenetic conclusions.

Annotated reviews are not available for download in order to protect the identity of reviewers who chose to remain anonymous.

---

## Round 0.2 · Minor Revisions

Thank you for implementing our suggestions. Your manuscript is as good as accepted, but there are some minor points which still need to be verified before publication, which cannot be changed anymore after acceptance.

Line 138: Do you mean “are” or rather “were”
Line 189: As far as I am aware all authors should be written in full in this case
Line 197: It is a bit dubious to refer to a publication which is still in preparation. I guess “Andreev, Coates & Sansom, unpublished” would be better, but at least some specifications of this analysis should be discussed in the caption (see previous reviews).
Line 315: I think “cyclomorial scale growth” needs to be defined somewhere (as this is hard to follow for non-experts) if not here than in the Discussion
Line 421-422: I agree, but it might be good to allude to the implications already here (or refer the to Discussion) as the implications first became apparent to me when reading the Discussion.
Line 712: “Histology of trunk scales” should be moved to the next line
Line 725: “Remarks” should be moved to the next line
Line 756: Please define “cyclomorial pattern” by referencing the original publication who coined the term or a subsequent reference adequately defining it.
Line 905-907: ok, the lower ends of the pulp canals are different in North American and Chinese mongolepids, but what could be the (potential) implications ? Do they belong to different taxa ? Do they grow differently?
Line 957-958: it kind of makes you wonder why there are these large stratigraphic gaps between these groups which are believed to be so closely related? Isn´t this also what´s dubbed the Devonian Nekton Revolution based on articulated material alone (Klug, C., Kröger, B., Kiessling, W., Mullins, G. L., Servais, T., Fryda, J., Korn, D., and Turner, S., 2010, The Devonian nekton revolution: Lethaia, v. 43, no. 4, p. 465-477).
Line 992-993: it is uncommon to refer to a manuscript which is still in preparation and not even uploaded on a preprint server. Please remove and cite as unpublished or upload it on a preprint server.
Figure 2: “spheritic dentine minerlisation” should be changed to “spheritic dentine mineralization”

I apologize for the inconvenience and looking forward to sent this into production once these points have been verified.

---

## Round 0.3 · accepted · Accept

Thank you for integrating these final suggestions. It has been a real pleasure to handle your manuscript. Your manuscript is now accepted for publication.